# Circulating Microparticles Are Differentially Increased in Lowlanders and Highlanders with High Altitude Induced Pulmonary Hypertension during the Cold Season

**DOI:** 10.3390/cells11192932

**Published:** 2022-09-20

**Authors:** Akylbek Sydykov, Aleksandar Petrovic, Abdirashit M. Maripov, Marija Gredic, Daniel Gerd Bermes, Nadira Kushubakova, Kubatbek Muratali Uulu, Christina Pilz, Meerim Cholponbaeva, Melis Duishobaev, Samatbek Satybaldyev, Nurgul Satieva, Argen Mamazhakypov, Meerim Sartmyrzaeva, Nazgul Omurzakova, Zhainagul Kerimbekova, Nursultan Baktybek, Cholpon Kulchoroeva, Oleg Pak, Lan Zhao, Norbert Weissmann, Sergey Avdeev, Leonid N. Maslov, Hossein Ardeschir Ghofrani, Ralph Theo Schermuly, Akpay S. Sarybaev, Djuro Kosanovic

**Affiliations:** 1Department of Internal Medicine, Member of the German Center for Lung Research (DZL), Justus Liebig University of Giessen, 35390 Giessen, Germany; 2Department of Mountain and Sleep Medicine and Pulmonary Hypertension, National Center of Cardiology and Internal Medicine, Bishkek 720040, Kyrgyzstan; 3Kyrgyz-Indian Mountain Biomedical Research Center, Bishkek 720040, Kyrgyzstan; 4Department of Medicine, Imperial College London, London SW7 2AZ, UK; 5Department of Pulmonology, Sechenov First Moscow State Medical University (Sechenov University), 119991 Moscow, Russia; 6Laboratory of Experimental Cardiology, Cardiology Research Institute, Tomsk National Research Medical Center, Russian Academy of Science, 634012 Tomsk, Russia

**Keywords:** high altitude, chronic cold exposure, pulmonary hypertension, echocardiography, microparticles

## Abstract

The role of microparticles (MPs) and cold in high altitude pulmonary hypertension (HAPH) remains unexplored. We investigated the impact of long-term cold exposure on the pulmonary circulation in lowlanders and high-altitude natives and the role of MPs. Pulmonary hemodynamics were evaluated using Doppler echocardiography at the end of the colder and warmer seasons. We further examined the miRNA content of MPs isolated from the study participants and studied their effects on human pulmonary artery smooth muscle (hPASMCs) and endothelial cells (hPAECs). Long-term exposure to cold environment was associated with an enhanced pulmonary artery pressure in highlanders. Plasma levels of CD62E-positive and CD68-positive MPs increased in response to cold in lowlanders and HAPH highlanders. The miRNA-210 expression contained in MPs differentially changed in response to cold in lowlanders and highlanders. MPs isolated from lowlanders and highlanders increased proliferation and reduced apoptosis of hPASMCs. Further, MPs isolated from warm-exposed HAPH highlanders and cold-exposed highlanders exerted the most pronounced effects on VEGF expression in hPAECs. We demonstrated that prolonged exposure to cold is associated with elevated pulmonary artery pressures, which are most pronounced in high-altitude residents. Further, the numbers of circulating MPs are differentially increased in lowlanders and HAPH highlanders during the colder season.

## 1. Introduction

Prolonged living at high altitudes causes a sustained rise in pulmonary artery pressure (PAP). It is general knowledge that pulmonary vasoconstriction and vascular remodeling due to chronic hypoxia exposure underlie high altitude pulmonary hypertension (HAPH) [1]. Yet, besides chronic hypoxia, cold is also an important environmental factor present in the high-altitude regions of the Earth [2]. Indeed, high altitude locations are characterized by extended periods with low temperatures. Observations of more frequent development of severe pulmonary hypertension (PH) with right ventricular failure in susceptible cattle after their exposure to high altitudes in cold seasons [3] suggested that cold might contribute to hypoxic PH. Subsequently, studies in large [4,5,6] and small animals [7,8,9,10,11,12] provided substantial evidence supporting the harmful effects of acute and chronic exposure to low temperatures on pulmonary circulation. Consistent with these experimental findings, human studies reported signs of PH in long-term residents and natives of the Canadian Arctic and Russian Northeast [13]. Furthermore, we have recently demonstrated that acute exposure to low temperatures leads to elevation of PAP in high-altitude residents of Kyrgyzstan [14]. However, no studies have addressed the impact of chronic cold exposure on pulmonary circulation in high-altitude dwellers yet.

In the past, we have reviewed and analyzed the knowledge about the influence of low temperatures on pulmonary circulation and existence of the cold-induced PH [13]. Based on the available literature, cold-induced PH is characterized by hypoxia and non-hypoxia driven pathological events, with enhanced inflammation, oxidative stress, and dysregulated phosphodiesterase system, which ultimately may cause the alteration of the pulmonary vascular cells and occurrence of the pulmonary vascular remodeling [13]. Furthermore, abnormally regulated inflammatory and pulmonary vascular cells may serve as an important source of microparticles (MPs).

MPs are shed cell membrane fragments released into the circulation to a certain extent in healthy individuals during different cellular processes [15]. Interestingly, both breathing hypoxic air [16,17] and high-altitude sojourn alter the MPs’ profile in healthy human subjects [18,19,20]. Further, augmented circulating MPs have been demonstrated in animal models of hypoxic PH [21]. Finally, we have recently revealed that inflammatory cell-derived MPs are increased in patients with almost all clinical forms of PH [22]. However, nothing is known regarding the role of MPs in the context of chronic HAPH in humans.

Overall, we hypothesize that endothelial and inflammatory cell-derived MPs are actively involved in the pathology of HAPH. Also, we suggest that a long-term cold environment as a regularly present factor together with hypoxic conditions at high-altitude locations, may provide additional effects on patients with HAPH and may further dysregulate the MPs profile. The objectives of the current study were, therefore, to explore the impact of chronic cold exposure on the pulmonary circulation in high-altitude residents and to investigate the role of MPs in HAPH.

## 2. Materials and Methods

### 2.1. High Altitude Expedition, Experimental Design, and Human Subjects

During 2017 we performed two expeditions in Kyrgyzstan, the first one at the end of the colder season (February–March) and the second one at the end of the warmer season (September–October) (Figure 1). Both expeditions were executed in the high-altitude village Sary-Mogol (approx. 3000 m) and Bishkek (approx. 760 m, lowland location). All human participants underwent clinical evaluation before inclusion: medical history, physical examination, blood pressure measurement, pulse oximetry, spirometry, 12-lead electrocardiography, echocardiography, and complete blood cell count. Exclusion criteria were major cardiovascular and respiratory diseases, active infections, pregnancy, and excessive erythrocytosis (hemoglobin concentration in females >19 g/dL, in males >21 g/dL). All participants in the study were Kyrgyz by origin and they agreed to participate during both seasons.

Based on the living location and echocardiography-derived tricuspid regurgitant systolic pressure gradient (TRG, mmHg), the study participants were divided into three groups: lowlanders (LA), highlanders controls (HA), and highlanders who developed pulmonary hypertension (HAPH) (Figure 1). For details with regard to the echocardiography and demographic profiles (age and gender ratio) of the participants (Table 1) please refer to our previously published work [14,23]. In addition, the peripheral blood was collected and drawn into citrated tubes from study participants during both seasons, following the plasma separation and isolation of MPs (Figure 1). Briefly, from the blood, the platelet-free plasma (PFP) was separated using the successive centrifugation protocol (500× *g*/15 min, followed by 10,000× *g*/5 min at room temperature), as we have described before [22]. In order to isolate the MPs, the second centrifugation step was performed using the PFP samples (15,000 × *g* (90 min), at 4 °C).

As we have already stated in our previously published studies derived from the expedition in Kyrgyzstan, written informed consents were obtained from all human subjects enrolled in the study [14,23]. The study was approved by the Ethics Committee of the National Centre for Cardiology and Internal Medicine, Bishkek, Kyrgyzstan, and by the Institutional Ethical Review Board of the Faculty of Medicine at Justus- Liebig University, Giessen, Germany. The study was executed in full agreement with the Declaration of Helsinki principles.

### 2.2. Flow Cytometry Characterization and Quantification of Endothelial and Different Inflammatory Cells-Derived MPs

Flow cytometry analysis of circulating MPs (CD68 (macrophages), CD3 (T-cells), and CD62E (endothelial cells)) in PFP was done as described previously by our group with subtle modifications [22]. Please find the details in the Appendix A.

### 2.3. Effects of Isolated MPs on Proliferation and Apoptosis of Human Pulmonary Artery Smooth Muscle Cells (PASMCs)

For assessment of apoptosis, Kinetic Apoptosis Kit (Abcam) and IncuCyte live cell imaging system were used. The proliferation of human PASMCs was determined by Cell Proliferation ELISA, 5-bromo-2′-deoxyuridine (BrdU) assay (Sigma Aldrich, St. Louis, MO, USA), following the manufacturer’s instructions. Please find the details in the Appendix A.

### 2.4. The Contents of Different MicroRNAs (miRNAs) in Isolated MPs: Quantitative Real-Time Polymerase Chain Reaction (qRT-PCR)

Isolated MPs were lyzed by the addition of 700 µL of QIAzol lysis buffer and total RNA was isolated using the miRNeasy Mini Kit (Qiagen), according to the manufacturer protocol. The amount of recovered RNA was quantified by Nanodrop (Thermo Fisher Scientific, Waltham, MA, USA). Expression levels of selected miRNAs: miRNA-210, miRNA-145, miRNA-17, and miRNA-133a-3p were assessed by qRT-PCR and TaqMan miRNA assays (Thermo Fisher Scientific). Please find the details in the Appendix A.

### 2.5. Effects of Isolated MPs on Human Pulmonary Artery Endothelial Cells (hPAECs): Tube Formation Assay

Primary hPAECs (PromoCell) were seeded into 0.1% gelatin (Sigma-Aldrich)-coated 35 mm culture dishes at 37 °C and pre-incubated with 10 µg/mL of isolated MPs or NaCl control for 24 h prior to trypsinization. hPAECs were re-suspended in Endothelial Cell Basal Medium (PromoCell) containing 0.2% FBS and 10 µg/mL of isolated MPs or NaCl control. A total of 12,500 cells were seeded in a 96-well plate coated with polymerized Matrigel Matrix (Corning) and transferred to IncuCyte live cell microscopy imaging system. Images obtained after 2 h were analyzed by ImageJ software and Angiogenesis Analyzer tool using various parameters, such as number of nodes, total length, and total segments length.

### 2.6. Effects of Isolated MPs on hPAECs: Western Blot

Primary hPAECs (PromoCell) passage 7 were cultured in Endothelial Cell Growth Medium (PromoCell) in the presence of 10 µg/mL of isolated MPs and incubated at 37 °C in a humidified atmosphere of 5% CO_2_ for 48 h prior to protein extraction. Western blot analyses were performed as described previously [24]. Anti-endothelial nitric oxide synthase (eNOS) (1:1000 dilution; BD Bioscience), anti-vascular endothelial growth factor (VEGF) (1:1000 dilution; Santa Cruz) or anti-β-actin (1:50,000 dilution; Sigma-Aldrich) were used as primary antibodies. Incubation with an appropriate secondary antibody coupled to horseradish-peroxidase (Promega) gave specific immune-reactive signals which were detected by enhanced chemiluminescence. Densitometry analyses were performed by Image Lab software (Bio-Rad).

### 2.7. Statistical Analysis

All results are presented as mean ± SEM. ROUT test was used for the identification of outliers. For statistical analysis of data, *t*-test with Welch’s correction and two-way ANOVA with Sidak´s multiple comparisons test were used. *p*-value < 0.05 was considered statistically significant.

## 3. Results

### 3.1. Long-Term Exposure to Cold Environment Enhances PAP in Kyrgyz Highlanders

First, we investigated the effects of cold exposure on pulmonary circulation in both Kyrgyz lowlanders (LA) and highlanders (HA). We compared intra-individually the changes in PAP, as estimated by TRG, between the colder and warmer seasons in 63 LA and 87 HA individuals. Interestingly, there was a trend toward higher TRG values in LA individuals during the cold season, as compared to those during the warm season (Figure 2). This effect was more profound in HA, and there was a significant increase in TRG values in individuals exposed to cold (Figure 2).

### 3.2. Effects of High Altitude and Cold on Circulating Levels of Different Endothelial and Inflammatory Cells-Derived MPs

The circulating levels of CD62E-positive MPs were comparable between the groups, except there was a significant increase in the number of these MPs in LA exposed to cold in comparison to the warm (Figure 3a). There were no changes among the groups in the case of CD3-positive MPs values (Figure 3b). However, CD68-positive MPs revealed the effects of cold exposure in the case of HAPH (Figure 3c). In addition, there was a significant increase in the number of these MPs in human individuals exposed to cold who developed PH, in comparison to the LA and HA controls exposed to cold (Figure 3c).

### 3.3. Effects of Isolated MPs on Human PASMCs

We have further investigated the potential effects of MPs isolated from the blood of human LA, HA, and HAPH exposed to the warmer or colder seasons on relevant cellular processes, such as the proliferation and apoptosis of human PASMCs. Our data revealed that isolated MPs from all experimental groups increased the proliferation of PASMCs (Figure 4a). Correspondingly, there was a significant reduction in apoptosis of PASMCs treated with isolated MPs from different experimental groups (Figure 4b).

### 3.4. Effects of High Altitude and Cold on the Content of Different miRNAs in Isolated MPs

With regard to the miRNA-145 and miRNA-133-3p, there were no changes in the expression among the experimental groups (Figure 5a,b). Furthermore, in the case of miRNA-17, the expression profiles were mostly comparable between the groups, except there was a significant upregulation of this miRNA in HAPH exposed to cold, as compared to the HA exposed to cold (Figure 5c). Finally, there were some interesting changes in expression profiles of miRNA-210 in various experimental groups. First, there was a significant downregulation of this miRNA in LA exposed to cold, in comparison to the LA exposed to warm (Figure 5d). Secondly, there was a significant upregulation of miRNA-210 in HA and HAPH exposed to cold, in comparison to the LA exposed to this condition (Figure 5d).

### 3.5. Effects of Isolated MPs on hPAECs

We have also investigated the potential effects of isolated MPs on other relevant cell type, such as hPAECs. Initially, we analyzed whether there were effects of isolated MPs on angiogenesis. However, there were no changes among the experimental groups, as evident from different parameters of angiogenesis, such as number of nodes, total length, and total segments length (Figure 6).

In addition, we have investigated the potential effects of MPs isolated from the blood of human LA, HA, and HAPH exposed to the warmer or colder season on expression profiles of VEGF and eNOS (Figure 7). Our results revealed a significant increase in expression of VEGF in HAPH as compared to the LA, both exposed to warm (Figure 7a). Furthermore, there was also upregulation in the expression of VEGF in HA compared to the LA, both exposed to cold (Figure 7a). With regard to the eNOS, there were no significant changes among the groups. However, there was a strong tendency toward reduced expression of eNOS in the HAPH exposed to cold, in comparison to the respective warm group and relevant LA and HA cold groups (Figure 7b).

## 4. Discussion

Overall, we have demonstrated that long-term exposure of humans to low temperatures is associated with PAP elevation. Interestingly, a recent study conducted on a large number of ambulatory patients in Israel showed that echocardiograms performed during Summer-Fall had a 19% lower chance to show systolic PAP > 40 mmHg compared with those done in Winter-Spring [25]. In our previous study, we revealed that acute exposure to cold leads to elevation of PAP in residents of Sary-Mogol village in South Kyrgyzstan [14]. It is important to note that the region, where Sary-Mogol is located, is characterized by significantly lower temperature conditions (average annual temperature (T) = 1.7 °C and average T during the colder season (November–March) = −8.9 °C) compared to other places at comparable altitudes [26]. Therefore, one could expect that lower temperature conditions might be associated with more severe alterations of the pulmonary circulation in high-altitude residents. However, there have been no direct comparisons of PAPs made between highlanders residing in places located at comparable altitudes but with different average annual temperatures.

MPs release is increased under stress conditions and disease. It has been suggested that a higher number of circulating MPs might be linked with different cardiovascular diseases [27]. In addition, circulating endothelial cell-derived MPs are potential biomarkers for pulmonary arterial hypertension (PAH), and even more, these vesicles may be actively involved in disease progression [28,29]. Dysregulation of endothelial MPs has been suggested as an early marker of vascular dysfunction [30]. Notably, different antigenic phenotypes of endothelial MPs allow identification of the nature of endothelial injury [31].

Hypoxia is known to activate endothelial cells [32] and CD62E-positive MPs have been shown to serve as a marker of endothelial activation [28]. However, we did not find any significant difference in levels of CD62E-positive MPs between lowlanders and highlanders. In line with our findings, the levels of CD62E-positive MPs were not significantly altered in healthy male individuals during a short-term stay at a moderate altitude of 2590 m [18]. In addition, a recent study demonstrated a slight decrease in CD62E-positive MPs compared to baseline sea level values in healthy volunteers at 3550 m [19]. Interestingly, the number of CD62E-positive MPs significantly increased at a further ascent to extreme altitudes [19]. These findings suggest, that endothelial cell activation probably does not occur at moderate altitudes and it appears to take place with severe hypoxia at extreme altitudes.

Several studies reported a positive correlation between endothelial cell-derived MPs and the severity of PAH [28,29]. Hypoxia is one of the main stimulators of endothelin-1 production [33]. High altitude exposure has been shown to be associated with an elevation in endothelin-1 plasma levels [34,35]. In subjects predisposed to high altitude pulmonary edema, endothelin-1 levels correlated with pulmonary hypertension severity [36,37,38]. In vitro studies demonstrated that in addition to its vasoregulatory properties, endothelin-1 stimulates the release of endothelial cell-derived microvesicles, which can then induce endothelial dysfunction [39]. In our study, the CD62E-positive MP levels were not different between healthy highlanders and HAPH highlanders. In support of our data, no correlation was found between CD62E-positive MP levels and hemodynamics severity in untreated PH patients [28]. In addition, CD62E-positive MP levels might depend on the etiology of the disease associated with PH [22,40]. Interestingly, we found that CD62E-positive MPs were elevated in lowlanders during the cold season but the reason for this remains unknown.

We have recently demonstrated enhanced levels of circulating CD3 cell-derived MPs in different forms of PH [22]. In contrast, we did not reveal any differences in CD3-positive MP levels among various experimental groups and conditions in our current study. These findings suggest that these MPs are probably involved neither in responses to cold nor in HAPH pathogenesis.

Similar to our previous report on other forms of PH [22], levels of CD68-positive MPs were not elevated in HAPH during the warm season. However, CD68-positive MPs were elevated in HAPH during the cold season. This finding is in line with the observation of increased infiltration of CD68-positive cells in the lungs of rats chronically exposed to cold [10]. Importantly, prolonged cold exposure in rats was associated with significant elevation of right ventricular systolic pressure, pulmonary vascular remodeling development, and right ventricular hypertrophy development [10]. Notably, inhibition of tumor necrosis factor-α abolished lung macrophage infiltration and attenuated right ventricular systolic pressure elevation, and improved pulmonary vascular remodeling [10] suggesting their potential role in the pathogenesis of cold-induced PH.

Accumulating evidence has suggested a role for MPs as carriers of various bioactive mediators, including miRNAs [15,41]. Interestingly, the majority of plasma miRNAs has been demonstrated to be associated with MPs [42]. Further, miRNA dysregulation has been identified as an important contributor to the pathogenesis of PH [43]. Various miRNAs have been shown to be involved in the pathogenesis of PH [44,45]. Thus, miRNAs have been implicated in the pulmonary vascular remodeling processes, including adventitial fibroblast migration; PASMCs proliferation, PAECs dysfunction, and inflammatory cell involvement [46,47]. In addition, a number of miRNAs is induced in response to hypoxia [48]. Of note, miRNA-210 has been identified as a major miRNA induced by hypoxia [49,50]. Further, a recent study evaluated plasma miRNA levels in healthy volunteers exposed to extremely high altitude and found that miRNA-17 and -190 significantly correlated with increased systolic PAP [51]. We did not reveal significant differences in miRNA levels between high-altitude natives and low-altitude residents. The reason for this finding is unclear, but the differential response of miRNAs to different types and severity of hypoxia may account for this. Indeed, a recent study found differences in serum miRNA-145 and -210 levels between patients with intermittent hypoxia and patients with chronic hypoxia [52].

Growing evidence suggests that miRNAs are critically involved in responses to adverse environmental conditions such as low temperature [53,54]. We did not find any differences in the levels of miRNA-145, -133-3p, and -17 between lowlanders and highlanders neither in winter nor in the summer season. Interestingly, the levels of MP-derived miRNA-210 in lowlanders decreased significantly during the winter season. This is in line with the finding of decreased serum miRNA-210 expression in rats exposed to cold [55]. Interestingly, attenuated white adipose tissue secreted exosomal miR-210 was linked to the activation of brown adipose tissue in high-altitude residents of Han Chinese descent [56]. In contrast, miRNA-210 levels in high-altitude residents during the winter season remained unchanged and were higher compared to those in lowlanders during the cold season. The reason for this is not clear, but may be related to the fact that winter in high-altitude areas is much colder and of significantly longer duration. Indeed, it has been suggested by other investigators that cold intensity and duration may change the expression of serum miRNAs [55].

MPs were suggested to affect the functions of PAECs and PASMCs. Circulating MPs isolated from hypoxia-induced PH rats have been demonstrated to affect the vascular tone and modulate vascular reactivity and influence neoangiogenesis. Cultured endothelial cells exposed to MPs isolated from rats with hypoxia-induced PH displayed decreased expression of eNOS and had lower nitric oxide production [21]. Additionally, ex vivo exposure of aortic and pulmonary artery rings to MPs obtained from pulmonary hypertensive rats exhibited impaired endothelium-dependent relaxation [21]. In our study, isolated MPs did not affect eNOS expression in human PAECs. However, there was a strong tendency towards the reduced expression of eNOS in the PAECs in the presence of MPs from HAPH highlanders exposed to cold, in comparison to the respective warm group and relevant low altitude and high-altitude cold groups.

VEGF is a potent endothelial cell-specific mitogen and permeability factor that is involved in angiogenesis [57]. Recent studies demonstrated the role of the interleukin-31/-33/hypoxia-inducible factor axis in the regulation of VEGF signaling [58,59]. There is substantial experimental evidence of increased production of VEGF in response to hypoxia [60]. However, human studies reported conflicting data regarding circulating VEGF-A levels in healthy sea level residents during short-term exposure to high altitudes [61,62]. Chronic exposure to hypoxia seems not to be associated with increased systemic VEGF levels in well-adapted Sherpa highlanders {Droma, 2020 #12125} [63]. Interestingly, patients with chronic mountain sickness exhibited higher VEGF expression compared to healthy Andean highlanders despite comparable peripheral oxygen saturation levels [64]. In our study, MPs isolated from highlanders with HAPH induced significantly higher expression of VEGF in PAECs compared to those isolated from lowlanders during the warmer season. Importantly, elevated circulating VEGF-A levels were reported in high altitude maladaptation conditions associated with PH [1,65], such as high altitude pulmonary edema [66] and chronic mountain sickness [67,68]. Indeed, highlanders with HAPH had significantly higher circulating VEGF-A levels compared to those without [67].

Cold exposure has been shown to stimulate VEGF expression in various tissues [69,70]. Interestingly, MPs isolated from healthy highlanders exposed to cold led to the upregulation of VEGF expression in PAECs compared to those from lowlanders exposed to cold suggesting a more pronounced effect by the combination of two factors [71]. Hypoxia and cold exposure have long been known to stimulate angiogenesis in skeletal muscles [72,73]. However, we did not reveal any impact of MPs on angiogenesis in human PAECs.

PASMCs in remodeled pulmonary vessels are characterized by over-proliferation and apoptosis resistance. Remarkably, extracellular vesicles isolated from monocrotaline-treated mice induced PH with increased muscularization of peripheral pulmonary arteries and right ventricular hypertrophy in healthy animals [74]. In accordance with these observations, MPs isolated from the blood of highlanders significantly increased proliferation and reduced apoptosis of PASMCs. Interestingly, in contrast to the previous experimental studies, MPs isolated from the blood of lowlanders similarly increased proliferation and reduced apoptosis of PASMCs. The reason for this is not clear, but might be related to the fact that lowlanders are actually also exposed to mild hypoxia (~800 m or equivalent oxygen fraction at sea level of 19%). The effects of MPs on human PASMCs were irrespective of the season and PH status.

## 5. Conclusions

Taken together, our study demonstrated that prolonged exposure to cold is associated with elevated PAPs, which is most pronounced in high-altitude residents. Further, the numbers of circulating MPs were differentially increased in lowlanders and HAPH highlanders during the colder season.

## Figures and Tables

**Figure 1 cells-11-02932-f001:**
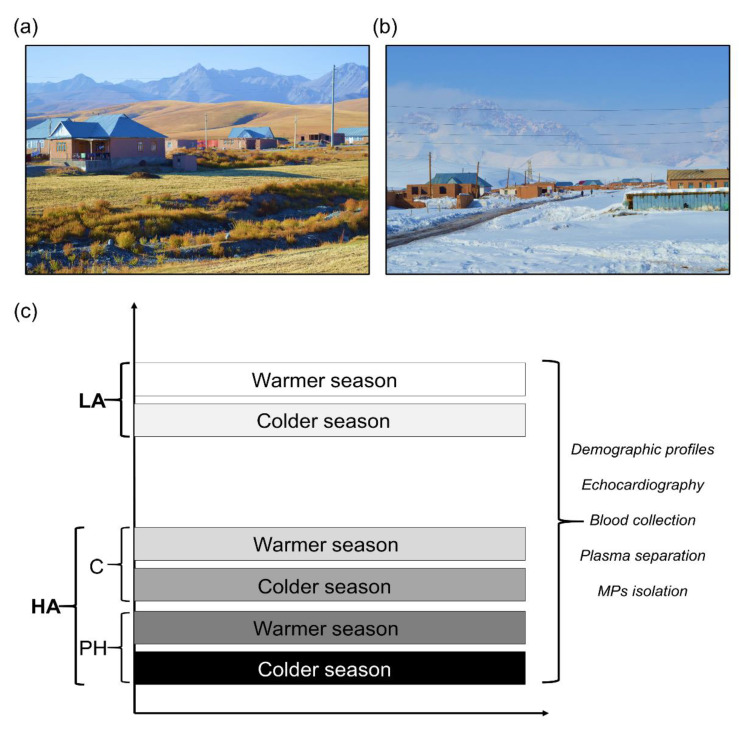
High altitude location in Kyrgyzstan and experimental design. The photographs from high altitude village Sary-Mogol (approx. 3000 m) during the late summer (**a**) warmer season and late winter (**b**) colder season are presented. The photographs may have been previously posted on different social media and used for the documentary film. The experimental design is depicted (**c**). LA—lowlanders, HA—highlanders, C—control, P—pulmonary hypertension, MPs—microparticles.

**Figure 2 cells-11-02932-f002:**
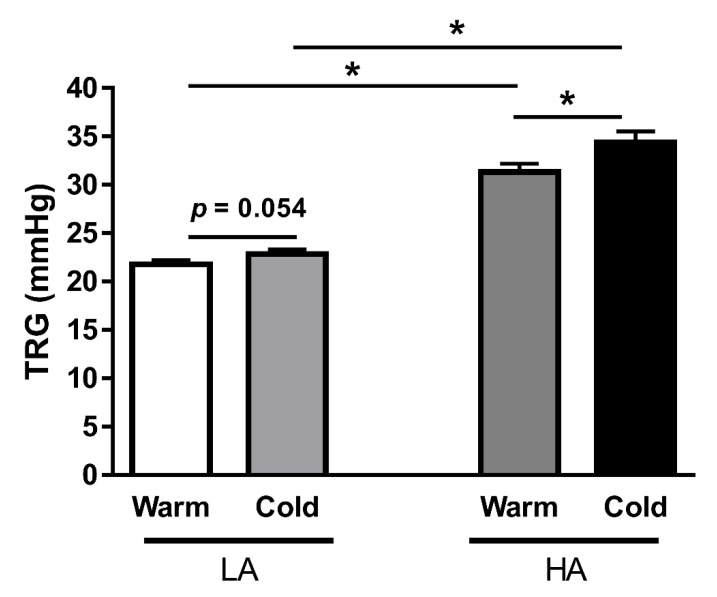
Effects of the long-term exposure to cold on pulmonary circulation in Kyrgyz lowlanders (LA) and highlanders (HA). TRG—tricuspid regurgitant systolic pressure gradient (in mmHg). Results are presented as mean ± SEM (n = 63–87). * *p* < 0.05 values are considered statistically significant.

**Figure 3 cells-11-02932-f003:**
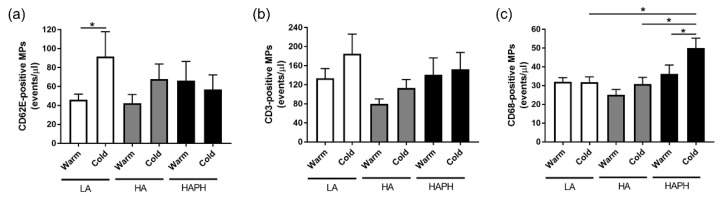
Effects of high altitude and cold on the circulating levels of different cells-derived microparticles (MPs) in Kyrgyz highlanders. (**a**–**c**) Flow cytometry characterization and quantification of different endothelial (CD62E) and inflammatory (T-cells (CD3) and macrophages (CD68)) cell-derived MPs are presented. LA—lowlanders, HA—highlanders, HAPH—highlanders with pulmonary hypertension. Results are presented as mean ± SEM (n = 10–12). * *p* < 0.05 values are considered statistically significant.

**Figure 4 cells-11-02932-f004:**
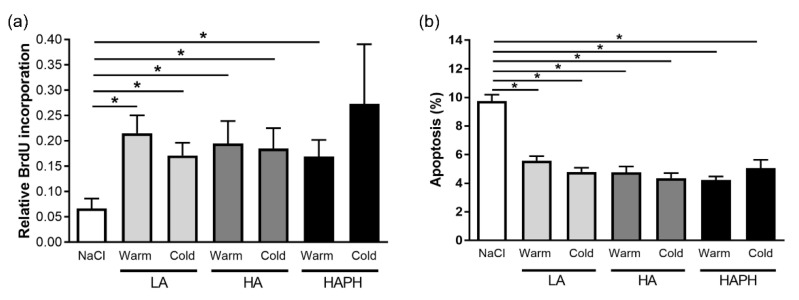
Effects of isolated microparticles (MPs) on proliferation and apoptosis of human pulmonary artery smooth muscle cells (PASMCs). Human PASMCs are exposed to isolated MPs and their effects on proliferation (**a**) and apoptosis (**b**) are shown. LA—lowlanders, HA—highlanders, HAPH—highlanders with pulmonary hypertension. Results are presented as mean ± SEM (n = 5–7). * *p* < 0.05 values are considered statistically significant.

**Figure 5 cells-11-02932-f005:**
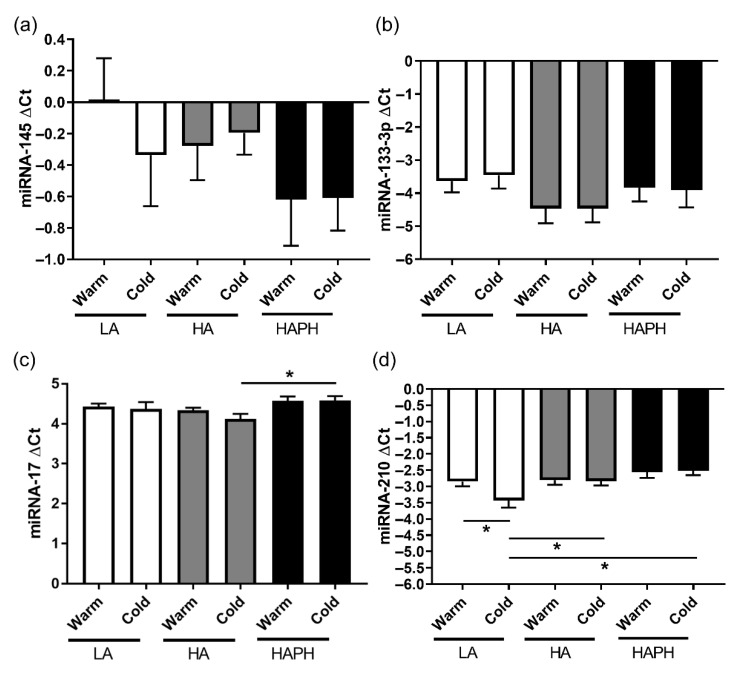
Effects of high altitude and cold on the content of different micro RNAs (miRNAs) in isolated microparticles (MPs). The characterization and quantification of different miRNAs in isolated MPs are performed by PCR and presented (**a**–**d**). LA—lowlanders, HA—highlanders, HAPH—highlanders with pulmonary hypertension. Results are presented as mean ± SEM (n = 11–12). * *p* < 0.05 values are considered statistically significant.

**Figure 6 cells-11-02932-f006:**
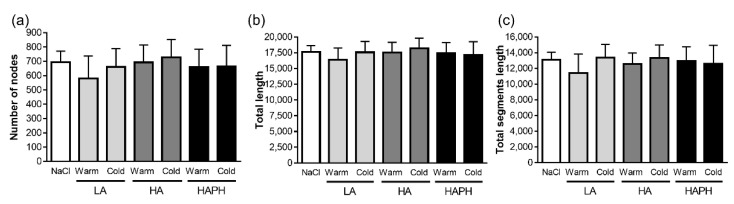
Effects of isolated microparticles (MPs) on human pulmonary artery endothelial cells (PAECs): angiogenesis. Human PAECs are exposed to isolated MPs and their effects on different parameters of angiogenesis (**a**–**c**) are presented. LA—lowlanders, HA—highlanders, HAPH—highlanders with pulmonary hypertension. Results are presented as mean ± SEM (n = 3).

**Figure 7 cells-11-02932-f007:**
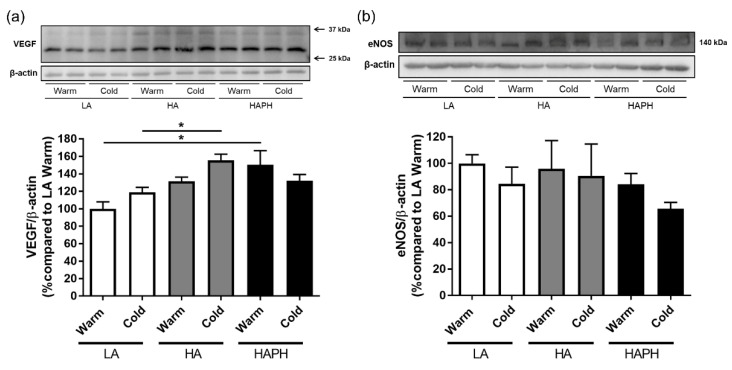
Effects of isolated microparticles (MPs) on human pulmonary artery endothelial cells (PAECs): expression profiles of vascular endothelial growth factor (VEGF) and endothelial nitric oxide synthase (eNOS). Human PAECs are exposed to isolated MPs and their effects on expression profiles of VEGF (**a**) and eNOS (**b**) are analyzed by Western Blot and presented. LA—lowlanders, HA—highlanders, HAPH—highlanders with pulmonary hypertension. Results are presented as mean ± SEM (n = 4). * *p* < 0.05 values are considered statistically significant.

**Table 1 cells-11-02932-t001:** Echocardiography-derived tricuspid regurgitant systolic pressure gradient (TRG) and demographic profiles in Kyrgyz subjects.

Experiment	Experimental Group	TRG (mmHg) Warm	TRG (mmHg) Cold	Age (Years)	f/m Ratio (%)	Associated Figure(s)
MPs numbers and miRNA content	LA (n = 12)	20.0 ± 0.5	23.2 ± 0.6 *	52.3 ± 2.1	42/58	3 and 5
	HA (n = 12)	21.1 ± 0.6	24.0 ± 0.9 *	37.6 ± 2.8	50/50	3 and 5
	HAPH (n = 12)	44.3 ± 1.0 ^§,$^	49.0 ± 1.1 *^,§,$^	50.8 ± 3.6	75/25	3 and 5
Effects of MPs on PASMCs	LA (n = 7)	19.3 ± 0.7	21.8 ± 0.4 *	51.9 ± 3.0	43/57	4
	HA (n = 7)	20.3 ± 1.0	22.2 ± 0.8 *	38.1 ± 4.5	43/57	4
	HAPH (n = 7)	45.2 ± 1.4 ^§,$^	50.1 ± 1.4 *^,§,$^	55.6 ± 5.2	71/29	4
Effects of MPs on PAECs: angiogenesis	LA (n = 3)	18.1 ± 0.6	21.0 ± 0.6	58.3 ± 2.7	67/33	6
	HA (n = 3)	20.4 ± 0.8	22.1 ± 0.9	44.0 ± 2.6	67/33	6
	HAPH (n = 3)	46.0 ± 3.3 ^§,$^	52.8 ± 1.8 *^,§,$^	52.7 ± 11.3	67/33	6
Effects of MPs on PAECs: Western Blot	LA (n = 4)	17.8 ± 0.5	21.1 ± 0.4 *	56.7 ± 2.5	50/50	7
	HA (n = 4)	19.3 ± 1.2	21.3 ± 1.0	41.0 ± 3.5	50/50	7
	HAPH (n = 4)	44.6 ± 2.7 ^§,$^	52.6 ± 1.3 *^,§,$^	53.7 ± 8.1	75/25	7

Data are presented as mean ± SEM. MPs—microparticles, miRNA—micro RNA, PASMCs—pulmonary artery smooth muscle cells, PAECs—pulmonary artery endothelial cells, LA—lowlanders, HA—highlanders, HAPH—highlanders with pulmonary hypertension, f—females, m—males, n—number of subjects. Two-way ANOVA with Sidak’s multiple comparisons test was used for statistical analysis. *p* < 0.05, * warm versus cold, ^§^—LA versus HAPH, ^$^—HA versus HAPH.

## Data Availability

All data are available from the corresponding author on reasonable request.

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
