# Peer review of "Circulating Microparticles Are Differentially Increased in Lowlanders and Highlanders with High Altitude Induced Pulmonary Hypertension during the Cold Season"

_cells, 2022, doi:10.3390/cells11192932_

Round 1
Reviewer 1 Report
The aim of the study was to investigate the impact of cold and altitude on the hemodynamic changes (pulmonary pressure) and concentration in microparticules (MPs) in blood in subjects pending on seasons and thus pending on temperature.
Pulmonary hemodynamic was evaluated using doppler echocardiography. Plasma levels of CD26/CD68 positive MPs was also evaluated.`The authors found an elevation in pulmonary pressure in highlanders while MPs in blood increased in response to cold exposure. They also evaluate the concentration in miRNA and the effects of the contains of MPs on human pulmonary artery smooth muscles (hPASMcs) and on endothelial cells (hPAECs). They found a different compostion of MPs between high and low landers. Cold exposure seems to stimlulate VEGF expression in hPAECs. They concluded that prolonged exposure to cold and/or to altitude is associated with elevated pulmonary artery pressure.
The study is of interest but needs some clarifications
Title ; to my point of view the authos did not prove that microparticules and cold are involved in high altitude induced pulmonary hypertension. The only found an association between the increase in MPs and cold exposure and found in vitro that the countains of these MPs may poarticpates into HHPP by the way of apoptosis and VEGF.
Thus i Suggest that the title could be : Circulating microparticles are increased (or are associated ) with cold during high altitude exposure : in vivo and in vitro approach
In this perspective the conclusion is wright
Introduction OK
Methods
Except for the subjects with High pulmonary blood pressure, were the other subjects free of any pathology that could interfere with the measurements, inflammatory process, chronic disease, asthma etc ? This should be precised
Table 11 is unclear
I suppose there was 3 groups
Low landers (LA)
Highlanders controls (HA)
And highlanders with HAPH
Please Can you assume the same separation into 3 clear groups in table 1
Results
3.1 It is specified that the authors compared TRG in 63 LA and 87 HA.
However in table 1 there is only 24 LA or HA ?
Please clarify : do you mean that there was a total of 150 subjects that have been evaluated on echocardiography and that among them only 78 (number of subjects in table 1) were studied for biological markers ? please clarify and arrange the table one.
Discussion
The work by Modesti PA et al highlithing the role of endothelin 1 should be cited
Circulation 2006 ; 114 :1410-6, because these authors were among the first to observe a relationship between altitude and pulmonary artery pressure and endothelin 1.
With warm clothes, subjects should not be too sensitive to the temperature dip, except for the inspiration of cold air. Do you think that it is by this way of cooling that part of your results could be explained?
Author Response
Q1. The aim of the study was to investigate the impact of cold and altitude on the hemodynamic changes (pulmonary pressure) and concentration in microparticules (MPs) in blood in subjects pending on seasons and thus pending on temperature. Pulmonary hemodynamic was evaluated using Doppler echocardiography. Plasma levels of CD26/CD68 positive MPs was also evaluated. The authors found an elevation in pulmonary pressure in highlanders while MPs in blood increased in response to cold exposure. They also evaluate the concentration in miRNA and the effects of the contains of MPs on human pulmonary artery smooth muscles (hPASMcs) and on endothelial cells (hPAECs). They found a different composition of MPs between high and lowlanders. Cold exposure seems to stimulate VEGF expression in hPAECs. They concluded that prolonged exposure to cold and/or to altitude is associated with elevated pulmonary artery pressure. The study is of interest but needs some clarifications
R1. We are very grateful to the reviewer for nice overview of our study and for supportive comments about our manuscript.
Q2. Title; to my point of view the authors did not prove that microparticules and cold are involved in high altitude induced pulmonary hypertension. The only found an association between the increase in MPs and cold exposure and found in vitro that the contains of these MPs may participate into HHPP by the way of apoptosis and VEGF. Thus I suggest that the title could be: Circulating microparticles are increased (or are associated) with cold during high altitude exposure : in vivo and in vitro approach. In this perspective the conclusion is wright
Introduction OK
R2. The reviewer’s point is well taken. As suggested by the reviewer, we modified the title. Now it reads as “Circulating microparticles are differentially increased in lowlanders and highlanders with high altitude induced pulmonary hypertension during the cold season”.
Q3 .Methods. Except for the subjects with High pulmonary blood pressure, were the other subjects free of any pathology that could interfere with the measurements, inflammatory process, chronic disease, asthma etc.? This should be precised
R3. We would like to thank the reviewer for this notion. All participants underwent functional evaluation before inclusion: thorough medical history, physical examination, blood pressure measurement, pulse oximetry, spirometry, complete blood cell count, 12-lead electrocardiography, and echocardiography. Exclusion criteria considered major cardiovascular and respiratory diseases, active infections, pregnancy, and excessive erythrocytosis (hemoglobin concentration in females >19 g/dL, in males >21 g/dL). Now, we have included this information in the revised manuscript.
Q3. Table 1 is unclear. I suppose there was 3 groups: Low landers (LA), Highlanders controls (HA), and highlanders with HAPH. Please can you assume the same separation into 3 clear groups in table 1
R4. We apologize for the confusion. We intended to provide TRG data and some other details for those subjects from these 3 groups, whose blood samples were used for the respective investigations. In the revised manuscript, we modified the table in order to make it clearer.
Q4. Results. 3.1 It is specified that the authors compared TRG in 63 LA and 87 HA. However in table 1 there is only 24 LA or HA? Please clarify: do you mean that there was a total of 150 subjects that have been evaluated on echocardiography and that among them only 78 (number of subjects in table 1) were studied for biological markers? Please clarify and arrange the table one.
R4. We apologize for the confusion. The TRG data presented in table 1 represent only data for those individuals whose blood samples were used for the respective in vitro investigations. In Figure 2, we presented TRG data for all 150 subjects. Although, we collected blood samples from all the study participants, not all the samples were used for in vitro experiments. Therefore, the number of subjects in table 1 does not correspond to that in Figure 2.
Q5. Discussion. The work by Modesti PA et al highlighting the role of endothelin 1 should be cited (Circulation 2006;114:1410-6), because these authors were among the first to observe a relationship between altitude and pulmonary artery pressure and endothelin 1.
R5. We thank the reviewer for this suggestion. In the revised manuscript, we included, the suggested work:
“Hypoxia is one of the main stimulators of endothelin-1 production (Wiley et al, 2001). High altitude exposure has been shown to be associated with an elevation in endothelin-1 plasma levels (Morganti et al, 1995; Goerre et al, 1995). In subjects predisposed to high altitude pulmonary edema, endothelin-1 levels correlated with pulmonary hypertension severity (Modesti et al, 2006; Droma et al, 1996; Sartori et al, 1999). In vitro studies demonstrated that in addition to its vasoregulatory properties, endothelin-1 stimulates release of endothelial cell-derived microvesicles, which can then induce endothelial dysfunction (Brewster et al, 2020).”
Q6. With warm clothes, subjects should not be too sensitive to the temperature dip, except for the inspiration of cold air. Do you think that it is by this way of cooling that part of your results could be explained?
R6.
In our opinion, this is a very important question. Unfortunately, the exact mechanisms of the effects of cold on pulmonary circulation remain poorly understood. Earlier studies demonstrated that cooling of the skin in cattle at thermoneutral ambient conditions increased pulmonary vascular resistance (McMurtry et al, 1975). Therefore, we share the reviewer’s opinion that warm clothes might ameliorate detrimental effects of cold. However, in regions with extreme cold temperatures, protection exerted by warm clothes might not be sufficient. Similarly, long-term cold exposure leads to development of persistent pulmonary hypertension even in various furbearers and wool animals (Watanabe et al, 2007; Crosswhite et al, 2013; Crosswhite et al, 2014; Van Bui et al, 1980; Banchero et al, 1987; Sakai et al, 2007; Chauca et al, 1976). In addition, cold can affect unprotected areas of the body. Indeed, it was shown that facial cooling produced an increase in pulmonary artery pressure and pulmonary vascular resistance in intensive care patients and in healthy high altitude residents (Giesbrecht et al, 1995). Finally, as reviewer suggested, pulmonary circulation can be affected by inspiration of cold air. Observations of clinical deterioration in pulmonary artery hypertension patients during cold weather and beneficial effects of protective measures like wearing scarfs indirectly support this idea.
Reviewer 2 Report
The manuscript is interesting and well written. I suggest to discuss the role of VEGF and IL-31/IL-33 axis in pathogenesis of endothelial dysfunction (see and add as references papers by Murdaca et al concerning these topics)
Author Response
Q1. The manuscript is interesting and well written. I suggest to discuss the role of VEGF and IL-31/IL-33 axis in pathogenesis of endothelial dysfunction (see and add as references papers by Murdaca et al concerning these topics)
R1. We are very grateful to the reviewer for nice overview of our study and for supportive comments about our manuscript. We would like to thank the reviewer for the suggestion. Professor Murdaca published excellent papers on the role of IL-31/IL-33 axis in the pathogenesis of allergic and rheumatologic diseases. We would be happy to include those papers in our manuscript. Unfortunately, this would affect the logical flow of the discussion. Instead we added the following sentence: “Recent studies demonstrated the role of interleukin-31/-33/hypoxia-inducible factor axis in the regulation of VEGF signaling (Liu et al, 2018; Borgia et al, 2022).”